# Lightweight Unsupervised Federated Learning with Pretrained Vision Language Model

## Abstract

Federated learning aims to tackle the "isolated data island" problem, where it trains a collective model from physically isolated clients while safeguarding the privacy of users' data. However, supervised federated learning necessitates that each client labels their data for training, which can be both time-consuming and resource-intensive, and may even be impractical for edge devices. Moreover, the training and transmission of deep models present challenges to the computation and communication capabilities of the clients. To address these two inherent challenges in supervised federated learning, we propose a novel lightweight unsupervised federated learning approach that leverages unlabeled data on each client to perform lightweight model training and communication by harnessing pretrained vision-language models, such as CLIP. By capitalizing on the zero-shot prediction capability and the well-trained image encoder of the pre-trained CLIP model, we have carefully crafted an efficient and resilient self-training approach. This method refines the initial zero-shot predicted pseudo-labels of unlabeled instances through the sole training of a linear classifier on top of the fixed image encoder. Additionally, to address data heterogeneity within each client, we propose a class-balanced text feature sampling strategy for generating synthetic instances in the feature space to support local training. Experiments are conducted on multiple benchmark datasets. The experimental results demonstrate that our proposed method greatly enhances model performance in comparison to CLIP's zero-shot predictions and even outperforms supervised federated learning benchmark methods given limited computational and communication overhead.

## 1 Introduction

Deep learning has achieved state-of-the-art performance across various benchmarks, primarily driven by the emergence of ultra-deep neural networks and the availability of centralized training data. While potential information hides in huge amount of personal or corporate data, learning from these isolated data islands poses a fundamental challenge in preserving privacy of user data. To address this challenge, federated learning (McMahan et al., 2017) was introduced as an interactive approach involving communication between a central server and individual clients. In this process, clients download initial model parameters from the server, update these parameters locally, and then upload the updated parameters back to the server. The server aggregates these updates and sends the aggregated parameters back to the clients. While FedAvg (McMahan et al., 2017) achieves rapid convergence when the data distribution among clients is homogeneous, heterogeneity in data distribution leads to biased local models and reduces the efficiency of federated learning(Luo et al., 2021). Subsequent research efforts have sought to enhance the training efficiency of heterogeneous federated learning, both at the client side (Li et al., 2020; Wang et al., 2020; Karimireddy et al., 2020; Li et al., 2021; Kim et al., 2022; Tan et al., 2022; Lee et al., 2022) and on the server side (Hsu et al., 2019; Reddi et al., 2021; Luo et al., 2021; Elgabli et al., 2022).

However, standard supervised federated learning faces two significant challenges. Firstly, it requires data annotation on every client, which is both time and resource-intensive. Secondly, updating deep models within the client and frequently transferring these models between the server and clients induce substantial computational and communication resources, particularly on edge devices such as mobile phones. Addressing these challenges has been the focus of only a few recent works. Some have explored semi-supervised federated learning, assuming that a portion of the data in each client

is labeled (Jeong et al., 2021; Diao et al., 2022). Lu et al. (2022) proposed federated learning from unlabeled data while under the strong assumption of known precise label frequencies on each client. Lin et al. (2022) proposed federated learning with positive and unlabeled data and assumed that each client labels only a portion of data from certain classes.

In this paper, we propose a novel lightweight unsupervised federated learning approach to simultaneously address the aforementioned annotation and resource demanding challenges. Our approach focuses on a setting where data annotation on each client is unnecessary, while restricting to lightweight model training on each client to accommodate computation and communication limitations. The contemplation of this learning approach is prompted by recent advancements in pretrained vision-language models, such as CLIP (Radford et al., 2021), which train both image and text encoders on large datasets of image-caption pairs and facilitate zero-shot predictions on downstream tasks by generating pairs of visual and textual features. While pretrained vision-language models can offer initial annotations through zero-shot prediction, achieving satisfactory or optimal model performance in the demanding context of lightweight and unsupervised federated learning still necessitates the development of novel methodologies.

To this end, we develop a novel method, Federated Self-Training with Class-Balanced Data Generation (FST-CBDG), to perform lightweight unsupervised federated learning by utilizing the text and image encoders of pretrained vision-language models. First, in the preparation stage, we generate the textual embeddings of all relevant classes using the pretrained text encoder on the server side and distribute them to participating clients along with the pretrained image encoder. Subsequently, in the federated learning stage, we form the prediction model by putting a lightweight linear classification layer on top of the pretrained image encoder, and conduct standard federated average learning solely on the linear layer. This learning scheme imposes minimal computational and communication overhead on each client. Additionally, the weight parameters of the linear classification layer can be conveniently initialized using the textual features of the corresponding class categories, which facilitates efficient federated learning by leveraging the zero-shot prediction capabilities of the pretrained vision-language model. Nevertheless, the crux of the matter is the efficient enhancement of initial models on each client within the constrained parameter space. Hence, we have carefully designed a self-training strategy aimed at improving the quality of predicted pseudo-labels and enhancing overall model performance. Moreover, to address the challenges and mitigate the negative impact of heterogeneous data distribution on local clients, we introduce a class-balanced data generation module to produce augmenting data from a Gaussian sampling model that leverages class-relevant text features. To evaluate our proposed approach, we conducted experiments on standard federated learning benchmarks under the "lightweight unsupervised federated learning" setting. The experimental results demonstrate that the proposed method achieves substantial improvements over CLIP's zero-shot prediction and even outperforms supervised federated learning benchmark methods given limited computational and communication overhead.

## 2 RELATED WORKS

**Federated learning**   Federated learning was introduced to address the challenge of training models based on isolated data islands. Majority studies focus on fully supervised federated learning settings, requiring every client has fully labeled data. The foundational FedAvg (McMahan et al., 2017) is a simple approach of averaging local model parameter updates on the server and sending them back to clients for further local updates. It demonstrates rapid convergence and approximation of centralized learning when client data adhered to an independently and identically distributed (i.i.d.) pattern. However, heterogeneity in data distribution among clients, which is common in real-world scenarios, introduces non-i.i.d. challenges, resulting in biased local models and slower model convergence. Subsequent research endeavors aimed to enhance heterogeneous federated learning. These approaches target both client and server-side improvements. FedProx (Li et al., 2020) enforces local parameter updates to stay close to the global model. FedNova (Wang et al., 2020) tackles the issue of objective inconsistency by employing a normalized averaging method. SCAFFOLD (Karimireddy et al., 2020) employs control variates to reduce variance in local updates. MOON (Li et al., 2021) corrects local updates by maximizing the agreement between local and global representations through contrastive learning. FedMLB (Kim et al., 2022) utilizes multi-level hybrid branching of modules from local and global models to generate multiple predictions, minimizing the Kullback-Leibler (KL) divergence of cross-branch predictions. FedNTD (Lee et al., 2022) gen-

erates outputs from global and local models, discarding logits belonging to the ground-truth class while minimizing the KL divergence between the modified predictions. FedBR (Guo et al., 2023b) reduces learning biases on local features and classifiers through mix-max optimization. FedDisco (Ye et al., 2023) aggregates local model parameters based on the discrepancy between local and global category distributions on the server. FedSMOO (Sun et al., 2023) adopts a dynamic regularizer to align local and global objectives and employs a global sharpness-aware minimization optimizer to find consistent flat minima. FedCLIP Lu et al. (2023) utilizes the pretrained CLIP model for federated learning while under the traditional supervised setting.

**Semi-Supervised Federated Learning** Recent works have relaxed the full supervision requirement and explored semi-supervised scenarios. FedMatch (Jeong et al., 2021) integrates federated learning and semi-supervised learning with an inter-client consistency loss. FedRGD (Zhang et al., 2021) employs consistency regularization loss and group normalization for local updates on the client side, along with a grouping-based model averaging method for aggregation on the server side. SemiFL (Diao et al., 2022) employs semi-supervised learning approaches for local updates and assumes extra labeled data on the server for aggregated model fine-tuning. Other works go even further to relax data annotation requirements. FedPU (Lin et al., 2022) assumes that each client labels only a portion of data from specific classes and uses positive and unlabeled learning methods for local updates. FedUL (Lu et al., 2022) introduces federated learning with only unlabeled data, but requires knowledge of precise label frequencies for each client.

**Pretrained Vision-language Models** Pretrained Vision-Language Models have gained popularity for their ability to learn image and text encoders from large image-text datasets. These models exhibit promising zero-shot prediction capabilities. CLIP (Radford et al., 2021) trains paired image and text encoders mainly used for image classification and retrieval. ALIGN (Jia et al., 2021) trains visual and language representations using noisy image and alt-text data. Subsequent models emphasize diverse tasks or expand the CLIP model. BLIP (Li et al., 2022) focuses on language-image pretraining for both vision-language understanding and generation with filtered captions. FLAVA (Singh et al., 2022) learns representations from paired and unpaired images and text, featuring multimodal and unimodal encoders. SimVLM (Wang et al., 2022) simplifies training complexity with large-scale weak supervision and a prefix language modeling objective. AltCLIP (Chen et al., 2023) extends CLIP's text encoder to a multilingual text encoder for multilingual understanding. Fashion-CLIP (Chia et al., 2022) and PLIP (Huang et al., 2023) fine-tune the CLIP model on special types of data. Recent research has harnessed such pretrained vision-language models, primarily CLIP, for various downstream applications. Menon & Vondrick (2023) leveraged large language models to generate descriptions for objects used in classification tasks, enhancing the zero-shot prediction capabilities of CLIP. Dunlap et al. (2023) employed CLIP to generate augmented domain-specific visual embedding for domain adaptation. Lüddecke & Ecker (2022) extended CLIP by incorporating a transformer-based decoder for semantic segmentation tasks. Gu et al. (2022) conducted knowledge distillation from a pretrained open-vocabulary image classification model into a two-stage detector for object detection. Guo et al. (2023a) adapted CLIP with prompt learning techniques for personalized supervised federated learning.

## 3 PROPOSED METHOD

In this section, we present the proposed method, Federated Self-Training with Class-Balanced Data Generation (FST-CBDG), for achieving lightweight unsupervised federated learning, where only unlabeled data, and limited computation and communication resources are available on each local client. The method centers on constructing a lightweight unsupervised federated learning framework by harnessing pretrained vision-language models, particularly CLIP, devising an effective self-training mechanism to improve noisy pseudo-labels and hence model performance through moving average soft label updates, and tackling the data imbalance and heterogeneity problem on local clients via class-balanced data generation. The framework of the proposed FST-CBDG method is presented in Figure 1. We elaborate this approach in subsequent subsections.

### 3.1 LIGHTWEIGHT UNSUPERVISED FEDERATED LEARNING FRAMEWORK

Deep classification models typically consist of a deep feature encoder that maps high-dimensional raw image data to high-level feature representations and a shallow classifier to make predictions

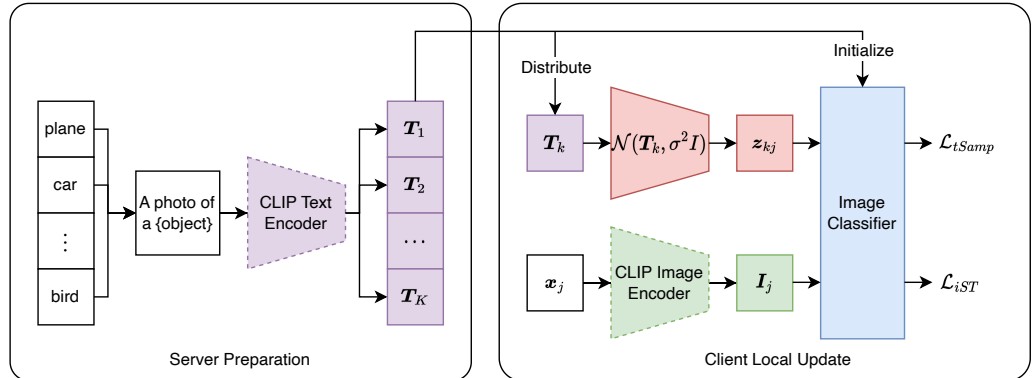

Figure 1: Framework of the proposed FST-CBDG method for lightweight unsupervised federated learning. In the server preparation stage, the CLIP image encoder and the categorical text features extracted using the CLIP text encoder are distributed to each client. During local training, extracted image features from the fixed CLIP image encoder are used for self-training of the linear classifier. Synthetic instances are generated in the feature space via class-balanced Gaussian sampling to address the data heterogeneity problem.

based on these high-level representations. However, training deep models on clients requires substantial labeled data and computational resources, and transmitting these models between clients and the server demands expensive communication bandwidth. To bypass such demanding training and communication requirements and realize lightweight unsupervised federated learning, we propose to initialize a federated learning framework by utilizing the recent pretrained vision-language models, particularly CLIP, for their impressive zero-shot transfer capabilities on downstream tasks.

CLIP trains image and text encoders using extensive datasets of image-caption pairs, offering well trained encoders that can extract visual and textual features in aligned feature spaces. With such aligned encoders, zero-shot image classification can be easily achieved by mapping extracted test image features based on cosine similarity to the text features extracted from sentences constructed from candidate category names. For unsupervised federated learning, we leverage CLIP's zero-shot prediction capability to prepare the federated learning model at the server side. Specifically, we first deploy CLIP's text encoder to extract textual features for the set of predefined class categories. For example, to obtain the textual feature vector for the class "plane", a sentence such like "a photo of a plane" can be input to CLIP's text encoder, resulting in the desired textual feature vector. Next, we form a prediction model by adding a linear classification layer on top of the pretrained and fixed CLIP image encoder, which produces a multi-class probabilistic classifier

$$f(\mathbf{z}; W, \mathbf{b}) = \text{softmax}(W\mathbf{z} + \mathbf{b}) \tag{1}$$

in the aligned feature space $\mathcal{Z}$. A linear classifier is chosen for two compelling reasons:

- Linear classifiers have significantly fewer parameters compared to full-fledged deep models, offering a lightweight training and transmitting mechanism for federated learning when fixing the pretrained CLIP image encoder.
- The weight parameters $W$ of the linear classifier can be initialized with textual features extracted from the CLIP text encoder for the predefined class categories, while setting $\mathbf{b} = 0$. Based on the zero-shot prediction capability of the CLIP model, this initialization not only provides the ability of predicting initial pseudo-labels for unlabeled data, but also can substantially enhance the convergence rate of the subsequent federated learning.

The textual features of the class categories and the initialized prediction model can be subsequently distributed to all the clients to produce the initial pseudo-labels on the unlabeled data and start the lightweight federated learning process: In each round, each client makes local updates on the linear classifier, which is then uploaded to the server for model aggregation; we adopt the simple average aggregation procedure of FedAvg. Therefore, we obtain a feasible initial framework for lightweight unsupervised federated learning.

### 3.2 RESILIENT SELF-TRAINING WITH EVOLVING PSEUDO-LABELS

Employing the pseudo-labels generated from CLIP model's zero-shot predictions as targets for federated learning, however, can often yield suboptimal results due to the low quality of these initial labels. An observation worth noting is that on benchmark datasets these initial predicted probabilities for each class are typically close to each other, and the CLIP zero-shot model tends to make low-confidence predictions on the unlabeled data. To empirically demonstrate the characteristics of the predicted probability vectors from the zero-shot CLIP model, we conducted an entropy analysis using a dataset of 1000 randomly sampled images from CIFAR-10 (Krizhevsky et al., 2009). To elaborate, let's denote an image as $\boldsymbol{x}_j$ and its extracted image features as $\boldsymbol{I}_j$. We also denote the text features for each class $k$ as $\boldsymbol{T}_k$, where $1 \leq k \leq K$, with $K$ being the total number of classes. The probability vector resulting from the CLIP zero-shot prediction for image $\boldsymbol{x}_j$ can be calculated as

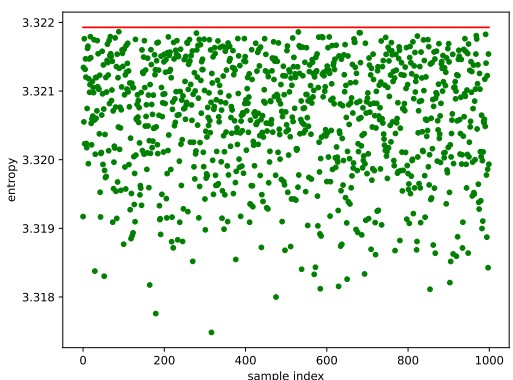

Figure 2: Entropy distribution of predicted probability vectors. Green dots represents the entropy for each sample and red line denotes the upper bound of the entropy ($\log 10 \approx 3.322$).

$$\boldsymbol{p}_j = [p_{j1}, \cdots, p_{jK}] = \text{softmax}([\boldsymbol{I}_j \cdot \boldsymbol{T}_1, \cdots, \boldsymbol{I}_j \cdot \boldsymbol{T}_K]). \tag{2}$$

The confidence level of the prediction can then be measured using the entropy value of the vector $\boldsymbol{p}_j$: $H(\boldsymbol{p}_j) = -\sum_{k=1}^K p_{jk} \log p_{jk}$. The upper bound for this entropy is $\log K$, which can only be reached when the predicted probability vector is a uniform vector. The results of this analysis are visualized in Figure 2 which shows the entropy values corresponding to the 1000 image samples as well as the upper bound for the entropy of a probability vector with $K = 10$ classes. From the figure, it is evident that the entropy values for all the sampled images are very close to the upper bound value of $\log(10)$. This observation demonstrates that the zero-shot CLIP model often produces probability vectors that are close to a uniform distribution across classes, resulting in low-confidence predictions.

To address the challenge posed by low-confidence initial pseudo-labels, we have devised a carefully crafted self-training method to progressively update and improve the pseudo-labels. It is evident that generating one-hot pseudo-labels from the low-confidence predictions during linear classifier training can often result in large errors and degrade the training process. Therefore, we opt for using soft pseudo-labels for self-training and update these labels using a moving average approach. In the $t$-th iteration, we use the following cross-entropy loss on *images* as the *Self-Training* objective for the linear classifier $f(\cdot)$ on each client:

$$\mathcal{L}_{iST} = -\mathbb{E}_{\boldsymbol{I}_j}[\boldsymbol{q}_j^t \cdot \log f(\boldsymbol{I}_j; W, \mathbf{b})] \tag{3}$$

As stated in the previous subsection, the weight matrix $W$ is initialized with the text features $\boldsymbol{T} = [\boldsymbol{T}_1, \cdots, \boldsymbol{T}_K]^\top$ and the bias vector $\mathbf{b}$ is initialized as $\mathbf{0}$ vector. The soft pseudo-labels, denoted as $\boldsymbol{q}_j$ are initially set to the CLIP zero-shot predicted probability vector, i.e. $\boldsymbol{q}_j^0 = \boldsymbol{p}_j$ and then updated with the model's prediction outputs. To obtain smooth and progressive updates of pseudo-labels and mitigate the risk of oscillations, we adopt the following weighted moving average update:

$$\boldsymbol{q}_j^t = \beta \boldsymbol{q}_j^{t-1} + (1 - \beta) f(\boldsymbol{I}_j; W, \mathbf{b}) \tag{4}$$

where $\beta$ is the hyper-parameter that controls the updating rate. This progressive update strategy can promptly incorporate the progress of the classifier training to improve the quality of pseudo-labels, while maintaining stability by accumulating the previous predictions.

### 3.3 CLASS-BALANCED DATA GENERATION

A significant challenge in federated learning arises from the non-i.i.d. data distribution across clients, which often results in class imbalances and introduces bias during local model training,

thereby diminishing the convergence rate of the global model. Regrettably, unsupervised federated learning exacerbates this situation since errors accumulated in the pseudo-labels further impede the convergence of the local models. Fortunately, there is a silver lining in the form of text features extracted from the CLIP text encoder for the relevant classes. As the CLIP model is trained using paired image-text data, the text features and image features pertaining to the same category exhibit a high degree of similarity, and the text feature vectors $\{\boldsymbol{T}_1, \cdots, \boldsymbol{T}_K\}$ can be regarded as class prototypes for the corresponding categories in the aligned image-text feature space $\mathcal{Z}$.

Using the $K$ text feature vectors—class prototype vectors—as additional labeled instances from their corresponding classes for training the local model however provides limited supervision and may lead to overfitting. Feature-level Gaussian augmentation has demonstrated effectiveness in recent works DeVries & Taylor (2017); Zhu et al. (2021). Motivated the clustering assumption that data belonging to the same class are usually close to each other in the high level feature space, we propose to model each class as a Gaussian distribution $\mathcal{N}(\boldsymbol{T}_k, \sigma^2 I)$ around the class prototype vector $\boldsymbol{T}_k$ in the feature space $\mathcal{Z}$, where $I$ denotes the identity matrix and $\sigma^2 I$ represents a diagonal covariance matrix. Then we can generate a set of synthetic instances for each $k$-th class in the feature space by randomly sampling feature vectors from the Gaussian distribution $\mathcal{N}(\boldsymbol{T}_k, \sigma^2 I)$, aiming to augment the pseudo-labeled training data and mitigate data heterogeneity and class imbalance. Specifically, we generate $n_k$ instances for each class $k$ as follows:

$$\{\boldsymbol{z}_{kj} \sim \mathcal{N}(\boldsymbol{T}_k, \sigma^2 I) | 1 \leq k \leq K, 1 \leq j \leq n_k\} \tag{5}$$

They can be used as labeled instances to help train the classifier by minimizing following cross-entropy loss:

$$\mathcal{L}_{tSamp} = -\sum_{k=1}^{K} \sum_{j=1}^{n_k} \mathbf{1}_k \cdot \log f(\boldsymbol{z}_{kj}; W, \boldsymbol{b}) \tag{6}$$

where $\mathbf{1}_k$ denotes the one-hot vector with a single 1 at the $k$-th entry.

To tackle the class imbalance problem at local clients, we further propose a class-balanced sampling strategy that generates more synthetic instances for the minority classes compared to the majority classes. To illustrate this, let's denote the number of images categorized into the $k$-th class based on the pseudo-labels ($k = \arg\max_{k'} \boldsymbol{q}_{jk'}^t$) on the considered client as $m_k$. The class-balanced sampling strategy determines the number of synthetic instances, $n_k$, based on the following balancing equation:

$$m_k + n_k = (1 + \gamma)m_{k^*}, \quad 1 \leq k \leq K \tag{7}$$

where $k^*$ denotes the class index with the largest number of predicted images on the considered client, such that $k^* = \arg\max_{k' \in \{1, \cdots, K\}} m_{k'}$; and $\gamma > 0$ controls the number of synthetic instances to be sampled for class $k^*$, specifically as $n_{k^*} = \gamma m_{k^*}$.

By utilizing all the pseudo-labeled real instances and generated synthetic instances, the linear classifier on each client is updated to minimize the following overall objective:

$$\min_{\boldsymbol{W}, \boldsymbol{b}} \quad \mathcal{L}_{iST} + \lambda \mathcal{L}_{tSamp} \tag{8}$$

where $\lambda$ is the trade-off parameter. The overall training algorithm for the proposed lighted unsupervised federated learning method, FST-CBDG, is presented in Algorithm 1 of Appendix A.

## 4 EXPERIMENTS

We conduct comprehensive experiments to assess the performance of the proposed method, Federated Self-Training with Class-Balanced Data Generation (FST-CBDG), under the lightweight unsupervised federated learning setting. Furthermore, we evaluate the proposed method in terms of computation and communication efficiency. Additional ablation study analyses contributions of individual components and examines the effects of certain hyper-parameters.

### 4.1 EXPERIMENTAL SETTINGS

**Datasets partition.** Following the experimental settings in Lee et al. (2022), we have conducted experiments on three datasets: CIFAR-10 (Krizhevsky et al., 2009), CIFAR-100 (Krizhevsky et al., 2009), and CINIC-10 (Darlow et al., 2018). To emulate the federated learning scenario, we divide

the data among $N = 100$ clients, ensuring no overlap. In each communication round, a random 10% of the clients participate in the federated training process. We have considered both homogeneous (i.i.d.) and heterogeneous (non-i.i.d.) data distribution settings. In the homogeneous setting, the data is evenly split and distributed to each client. In contrast, the heterogeneous setting involves the use of two widely recognized partition methods: *Sharding* and Latent Dirichlet Allocation (*LDA*). *Sharding* involves sorting the data based on the labels and then dividing them into $Ns$ shards where $s$ represents the number of shards per client. Each client subsequently randomly selects $s$ shards without replacement to constitute its local data. The parameter $s$ controls the data heterogeneity, with smaller values of $s$ leading to higher levels of data heterogeneity. We conducted experiments on all three datasets using various values: CIFAR-10 ($s$ values of 2, 3, 5, and 10), CIFAR-100 ($s$ value of 10) and CINIC-10 ($s$ value of 2). On the other hand, the *LDA* method partitions each class of data to each client according to a Dirichlet distribution with a parameter $\alpha$. For any given class $k$, each client $i$ randomly samples a proportion $p_{ki}$ of the data belonging to class $k$, where $p_{ki} \sim Dir(\alpha)$ and $\sum_{i=1}^{N} p_{ki} = 1$. The parameter $\alpha$ controls the data heterogeneity within each client, with smaller values of $\alpha$ indicating more severe data heterogeneity. In our experiments, we used various $\alpha$ values for the three datasets, CIFAR-10 ($\alpha$ values of 0.05, 0.1, 0.3, 0.5), CIFAR-100 ($\alpha$ value of 0.1) and CINIC-10 ($\alpha$ value of 0.1). It's important to note that in the context of unsupervised federated learning, all data within each client are unlabeled.

**Implementation details.** CLIP offers various pretrained image encoders with different model architectures. Specifically, we chose a simple variant, 'RN50', in which the global average pooling layer of the original ResNet-50 model is replaced with an attention pooling mechanism (Radford et al., 2021). The pretrained text encoder is based on a modified Transformer model (Vaswani et al., 2017). The linear classifier has a input size of 1024, which matches the the output size of the CLIP image encoder. We optimized the linear classifier using mini-batch Stochastic Gradient Descent (SGD) with a learning rate of 0.01, a momentum of 0.9 and a weight decay of $10^{-5}$. For the proposed method, we set the moving average parameter of the pseudo-label updating $\beta$, to 0.9, and the class-balanced sampling parameter $\gamma$ to 0. The trade-off parameter between the self-training and text sampling losses $\lambda$ was set to 1. Given the lightweight setting, we limited the number of communication rounds to 10, and each client performed 1 local update epoch for each round.

**Baselines.** In our experiments, we compared our proposed method, FST-CBDG, with two baseline approaches and two representative supervised federated learning methods. CLIP-ZS represents using the pretrained CLIP model to make zero-shot prediction on the testing data. CLIP-FC-Centralized denotes that we train a linear classifier based on the fixed CLIP image encoder with SGD optimizer in a centralized manner. The classifier was trained on all the training data with labels and evaluated on the testing data. As comparison, FedAvg (McMahan et al., 2017) and FedNTD (Lee et al., 2022) are adapted to train a linear classifier based on the fixed CLIP image encoder (RN50 variant) with labeled training data in each client. Our proposed method, FST-CBDG, differs from the above methods as it trains a linear classifier in a federated manner, but all the data in each client are unlabeled. This introduces a more challenging setting compared to the supervised federated learning methods.

## 4.2 COMPARISON RESULTS

### 4.2.1 PERFORMANCE ON HOMOGENEOUS DATA

In our evaluation under the homogeneous data distribution setting, we compared the performance of the proposed FST-CBDG method with several baselines, including CLIP-ZS, CLIP-FC-Centralized, FedAvg, and FedNTD, on three datasets: CIFAR-10, CIFAR-100, and CINIC-10. Here are the key findings from the results in Table 1. CLIP-ZS achieves decent performance on all three datasets. It serves as a strong baseline, leveraging the pretrained CLIP model's transfer capabilities. CLIP-FC-Centralized, which trains a linear classifier using the fixed CLIP image encoder and labeled training data in a centralized manner, significantly improves performance compared to CLIP-ZS. FedAvg and FedNTD, these supervised federated learning methods, which also train linear classifiers based on the fixed CLIP image encoder but with labeled data, outperform CLIP-ZS predictions. Our proposed method, FST-CBDG, which operates in a federated manner with unlabeled data, outperforms CLIP-ZS by a significant margin on all three datasets. It even surpasses the performance of the supervised federated learning methods, FedAvg and FedNTD. Notably, FST-CBDG achieves performance that is close to the centralized and supervised baseline, CLIP-FC-Centralized.

Table 1: Testing accuracy (%) under homogeneous data distribution.

| Methods | Supervised | CIFAR-10 | CIFAR-100 | CINIC-10 |
|---|---|---|---|---|
| CLIP-ZS | - | 68.7 | 39.0 | 63.2 |
| CLIP-FC-Centralized | ✓ | 77.5 | 42.9 | 70.4 |
| FedAvg | ✓ | 73.3 | 37.8 | 66.0 |
| FedNTD | ✓ | 72.8 | 39.8 | 66.2 |
| FST-CBDG (ours) | ✗ | 74.0 | 43.2 | 66.3 |

Table 2: Testing accuracy (%) under heterogeneous data distribution.

| NIID Partition Strategy: Sharding | | | | | | | |
|---|---|---|---|---|---|---|---|
| Methods | Supervised | CIFAR-10 | | | | CIFAR-100 | CINIC-10 |
| | | $s=2$ | $s=3$ | $s=5$ | $s=10$ | $s=10$ | $s=2$ |
| CLIP-ZS | - | 68.7 | | | | 39.0 | 63.2 |
| CLIP-FC-Centralized | ✓ | 77.5 | | | | 42.9 | 70.4 |
| FedAvg | ✓ | 32.3 | 42.0 | 43.5 | 47.7 | 34.1 | 30.9 |
| FedNTD | ✓ | 42.0 | 64.1 | 47.6 | 55.6 | 26.6 | 35.8 |
| FST-CBDG (ours) | ✗ | 72.0 | 72.8 | 73.6 | 73.2 | 43.3 | 65.9 |
| NIID Partition Strategy: LDA. | | | | | | | |
| Method | Supervised | CIFAR-10 | | | | CIFAR-100 | CINIC-10 |
| | | $\alpha=0.05$ | $\alpha=0.1$ | $\alpha=0.3$ | $\alpha=0.5$ | $\alpha=0.1$ | $\alpha=0.1$ |
| FedAvg | ✓ | 20.1 | 32.4 | 41.9 | 45.1 | 16.4 | 29.1 |
| FedNTD | ✓ | 26.6 | 28.2 | 37.1 | 52.9 | 15.9 | 26.6 |
| FST-CBDG (ours) | ✗ | 71.5 | 71.9 | 72.2 | 72.4 | 43.1 | 65.0 |

### 4.2.2 PERFORMANCE ON HETEROGENEOUS DATA

In our evaluation under the more challenging setting of heterogeneous data distribution, we considered two different data construction strategies: *Sharding* and *LDA*. Here are the key findings from the results in Table 2. Compared with the CLIP-ZS baseline, our method FST-CBDG consistently enhances model performance across all three datasets in the challenging heterogeneous setting. FST-CBDG also outperforms the supervised federated learning methods across different datasets and heterogeneous data partition strategies even though our method trains the model without labels. It's interesting to notice that the supervised federated learning methods fail under the lightweight heterogeneous federated learning setting even though labeled data are given. With limited communication rounds and local update epochs, FedAvg and FedNTD cannot preserve the initial performance of the CLIP zero-shot predictions. On the one hand, the strong supervision from the labeled data introduce negatives transferring effect to the linear model. On the other hand, data heterogeneity in the local client leads to biased local models thus biased aggregated global model while FedAvg and FedNTD failed to address this under the lightweight setting. However, our method FST-CBDG not only consistently improve the performance starting from the CLIP zero-shot prediction through the proposed resilient self-training method, but also reduce the influence of data heterogeneity by sampling synthetic instances.

### 4.2.3 COMPUTATION AND COMMUNICATION EFFICIENCY

Figure 3 displays the testing accuracy curves concerning the communication rounds for all three datasets. FST-CBDG exhibits rapid convergence in both homogeneous and heterogeneous data distribution settings. The curves begin at the accuracy level of the CLIP zero-shot prediction, and while the two comparison methods fail to maintain this initial accuracy, FST-CBDG consistently improves accuracy and achieves near-optimal performance within a few communication rounds: CIFAR-10 (1 round), CIFAR-100 (6 rounds), and CINIC-10 (1 round). This indicates that the proposed method greatly reduces the computation and communication requirements for the client devices.

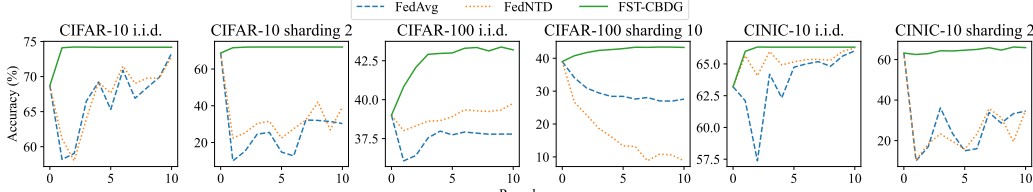

Figure 3: Curves of the testing accuracy (%) w.r.t. communication rounds for the proposed method FST-CBDG and the two comparison methods, FedAvg and FedNTD under homogeneous (i.i.d.) and heterogeneous (Sharding) data distribution.

### 4.2.4 ABLATION STUDY

**Components.** In our ablation study, we examined the effects of the self-training loss and synthetic instance sampling loss in our proposed method. The results, as presented in Table 3, reveal that using either the self-training loss or text sampling loss alone does not lead to performance improvements over the CLIP-ZS baseline. We also conducted an experiment to assess centralized training using the text sampling loss, but it also failed to produce improvements. However, when both losses are combined, the results demonstrate significant improvements over the baseline, underscoring the necessity of both components for our approach.

**Sampling strategy.** We conducted experiments to assess the effectiveness of our proposed class-balanced sampling strategy, as outlined in Equation 7, by comparing it to another sampling strategy, equal sampling. The equal sampling strategy involves sampling the same number of synthetic instances for each class, irrespective of the number of images per class in the local client. The results, presented in Table 4, demonstrate the superiority of our proposed class-balanced sampling strategy over the equal sampling approach. Our class-balanced sampling strategy consistently outperforms equal sampling by a significant margin across all datasets, regardless of whether the data distribution is homogeneous or heterogeneous. This highlights the effectiveness of our carefully designed sampling strategy in improving model performance.

Table 3: Ablation study to evaluate each component. Test accuracy (%) on each dataset under i.i.d. and non-i.i.d. (Sharding) data distribution.

| Method | CIFAR-10 | | CIFAR-100 | |
|---|---|---|---|---|
| | $s = 2$ | i.i.d. | $s = 10$ | i.i.d. |
| CLIP-ZS | 68.7 | | 39.0 | |
| $\mathcal{L}_{iST}$ | 68.9 | 68.2 | 37.5 | 37.5 |
| $\mathcal{L}_{tSamp}$ | 68.9 | 65.7 | 37.3 | 35.5 |
| $\mathcal{L}_{tSamp}$ (Centralized) | 69.4 | 69.4 | 39.1 | 39.1 |
| $\mathcal{L}_{iST} + \mathcal{L}_{tSamp}$ | 72.0 | 74.0 | 43.3 | 43.2 |

Table 4: Ablation study to evaluate sampling strategy. Test accuracy (%) on each dataset under i.i.d. and non-i.i.d. (Sharding) data distribution.

| Sampling strategy | CIFAR-10 | | CIFAR-100 | |
|---|---|---|---|---|
| | s=10 | i.i.d. | s=10 | i.i.d. |
| Equal sampling | 68.6 | 68.6 | 37.4 | 37.4 |
| Balanced sampling | 73.2 | 74.0 | 43.3 | 43.2 |

## 5 CONCLUSION

In this paper, we proposed a novel lightweight unsupervised federated learning approach, FST-CBDG, to alleviate the computational and communication costs, as well as the high data annotation requirements typically associated with standard federated learning for deep models. By capitalizing on the petrained visual-language model CLIP, the proposed method devises an efficient and resilient self-training approach to progressively refine the initial pseudo-labels produced by CLIP and learn a linear classifier on top of the fixed CLIP image encoder. Additionally, we propose a class-balanced synthetic instance generation method based on the class prototypes produced by the CLIP text encoder to address data heterogeneity within each client. The experimental results on multiple datasets demonstrate that the proposed method greatly improves model performance in comparison to CLIP's zero-shot predictions and outperforms supervised federated learning benchmark methods given limited computational and communication overhead.

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

# A  ALGORITHM

---

**Algorithm 1:** Federated Self-Training with Class-Balanced Data Generation (FST-CBDG)

---

**Input** : Number of classes $K$, total number of communication rounds $R$, learning rate $\eta$.

    `/* Server preparation                                                 */`

1  **for** *each class $k \leftarrow 1$* **to** $K$ **do**

2     |   Obtain class name {object} and construct a sentence: a photo of a {object}.

3     |   Extract text feature vector $\boldsymbol{T}_k$ using the CLIP text encoder from the sentence above.

4  **end**

5  Initialize the parameters of the linear classifier $\boldsymbol{W} = [\boldsymbol{T}_1, \cdots, \boldsymbol{T}_K]^\top$ and $\boldsymbol{b} = \boldsymbol{0}$.

6  Distribute the text features $\{T_k\}_{k=1}^K$ and CLIP image encoder to each client.

    `/* Training starts                                                     */`

7  **for** *each round $r \leftarrow 1$* **to** $R$ **do**

8     |   Server samples participated clients for training in this round.

    |   `/* Local update                                                   */`

9     |   **for** *each client $c$* **do**

10     |   |   Download model parameters $\boldsymbol{W}_c^r$ and $\boldsymbol{b}_c^r$ to local machine.

11     |   |   **for** *each iteration* **do**

12     |   |   |   Extract image feature vectors $\boldsymbol{I}_j$ for each image $\boldsymbol{x}_j$ using the CLIP image encoder.

13     |   |   |   Update soft pseudo labels for each image $\boldsymbol{x}_j$ according to Equation (4).

14     |   |   |   Calculate self-training loss $\mathcal{L}_{iST}$ according to Equation (3).

15     |   |   |   Sample synthetic instances according to Equation (5) and (7).

16     |   |   |   Calculate loss based on sampled synthetic instances $\mathcal{L}_{tSamp}$ via Equation (6).

17     |   |   |   Update model parameters $\boldsymbol{W}_c^r \leftarrow \boldsymbol{W}_c^r - \eta \nabla_{\boldsymbol{W}_c^r}(\mathcal{L}_{iST} + \lambda \mathcal{L}_{tSamp})$ and
    |   |   |     $\boldsymbol{b}_c^r \leftarrow \boldsymbol{b}_c^r - \eta \nabla_{\boldsymbol{b}_c^r}(\mathcal{L}_{iST} + \lambda \mathcal{L}_{tSamp})$

18     |   |   **end**

19     |   |   Upload updated model parameters $\boldsymbol{W}_c^r$ and $\boldsymbol{b}_c^r$ to the server.

20     |   **end**

    |   `/* Model aggregation in the server                                */`

21     |   $\boldsymbol{W}^{r+1} \leftarrow \mathbb{E}_c[\boldsymbol{W}_c^r]$ and $\boldsymbol{b}^{r+1} \leftarrow \mathbb{E}_c[\boldsymbol{b}_c^r]$

22  **end**

---

