# OpenReview forum: "Lightweight Unsupervised Federated Learning with Pretrained Vision Language Model"
_ICLR.cc/2024/Conference — Submitted to ICLR 2024_

### Official Review · Reviewer_91xa · 2023-10-29

**Soundness:** 2 fair
**Presentation:** 3 good
**Contribution:** 2 fair
**Rating:** 3
**Confidence:** 4

**Summary:**

This paper uses the CLIP image and text encoder for unsupervised learning in FL. Specifically, it uses the image encoder and text features extracted in the server in the client for training. Besides, it also generates data samples on clients to mitigate data imbalance problems.

**Strengths:**

- The idea is interesting and the Figure 1 illustrates the idea well.
- The paper the generally well-written and easy to follow.
- The proposed method achieves significantly better performance than compared counterparts, especially under heterogeneous data distribution.
- The proposed method only requires a few rounds of communication.

**Weaknesses:**

- Some important experimental details are missing. For example, the model architecture used for training the other methods.
- The paper mentions computation efficiency in Section 4.2.3, but it seems that the computation saving is mainly from the reduction of the communication round. The paper seems to focus on cross-device FL. It would be useful to further investigate whether the device could fit in the model size of CLIP and has enough memory to use it for inference.
- The compared methods are somewhat weak baselines. Some important unsupervised FL baselines and methods are not discussed or compared in the paper, e.g., [1][2][3][4]
    - [1] Collaborative Unsupervised Visual Representation Learning from Decentralized Data. ICCV’21
    - [2] Divergence-aware Federated Self-Supervised Learning. ICLR’22.
    - [3] Orchestra: Unsupervised Federated Learning via Globally Consistent Clustering. ICML’22
    - [4] MocoSFL: Enabling Cross-client Collaborative Self-supervised Learning. ICLR’23
- Some papers mentioning adopting CLIP to FL are not discussed. e.g. [5][6]
    - [5] Fedclip: Fast generalization and personalization for clip in federated learning.
    - [6] When Foundation Model Meets Federated Learning: Motivations, Challenges, and Future Directions.

**Questions:**

- What is the impact of using different types of backbone?
- What is the impact of training for more local epochs in each round?

---

> ### Author Response · Authors · 2023-11-21
> **Reply from Authors**
>
> We thank the reviewer for dedicating time and effort to review our paper.
>
> **1. Experimental details**
>
> As outlined in subsection 4.1 of the experimental settings, it is essential to note that all the baselines adopt the same model architecture as the proposed method to ensure a fair comparison. Each baseline utilizes the pretrained CLIP image encoder (RN-50 variant) as the fixed encoder and subsequently trains a linear layer classifier. To enhance clarity, we have revised the descriptions in the baseline section accordingly.
>
> **2. Computation efficiency**
>
> We kindly direct the reviewer to the authors' response to the second question posed by reviewer b2QK for further information and clarification.
>
> **3. Fitting CLIP for inference in local clients**
>
> Given the constrained computation resources in local clients, our proposed lightweight framework adopts a strategy where textual embeddings are generated on the server side and then distributed to each client. This approach eliminates the need for repeated inference from the CLIP text encoder on each client. To further alleviate the local computation requirements, our method utilizes the smallest variant of the CLIP image encoder, namely RN50. Devices capable of accommodating ResNet-50 for inference can similarly support the model in our proposed method.
>
> **4. Other works on self-supervised federated representation learning**
>
> This paper tackles the novel challenge of unsupervised federated learning and presents a lightweight framework designed to address it. The framework incorporates the pretrained CLIP image encoder, fixing it to minimize the computation and communication demands on local clients. The listed works, however, focus on self-supervised federated representation learning. In contrast to our approach, these methods propose techniques to train the image encoder in a federated manner. Consequently, they are not directly relevant to the problem addressed by this paper.
>
> **5. Other papers adopting CLIP to FL**
>
> We thank the reviewers for providing the related works and have included them in the related works part.
> It's essential to highlight that FedCLIP similarly incorporates the pretrained CLIP model into federated learning. However, it is designed to address the traditional supervised federated learning setting where labeled data are available in the clients. In contrast, our method focuses on the unsupervised federated learning setting, where only unlabeled data is available in the clients. This distinction underscores the unique contribution and applicability of our approach in scenarios with exclusively unlabeled client data.
>
> **6. Impact of difference encoder backbones**
>
> The CLIP model offers a range of image encoder variants, each striking a balance between zero-shot prediction ability and model size. In the context of the lightweight unsupervised federated learning setting and with the aim of minimizing computational requirements on local clients, we opted for the smallest variant, RN-50, as the fixed image encoder. It's noteworthy that while the proposed method's performance benefits from the CLIP zero-shot prediction model, incorporating larger variants of the CLIP image encoder can further enhance performance, starting from the corresponding zero-shot performance of the selected image encoder. However, this improvement comes at the cost of increased computational demands on the local clients.
>
> **7. Impact of more local update epochs**
>
> We conducted an additional ablation study to explore the influence of varying the number of local update epochs. The results for the two datasets, CIFAR-10 and CIFAR-100, under both homogeneous and heterogeneous settings are presented in the following table. Increasing the number of local update epochs did not lead to a notable improvement in the performance of the proposed method. The overall best performance was achieved with only 1 local update epoch. This observation underscores the lightweight property of the proposed method. Moreover, it eliminates the necessity for tuning this hyper-parameter, simplifying the implementation. This phenomenon can be attributed to the high entropy property of CLIP zero-shot prediction, emphasizing the need for stable local updating.
>
> | # epochs | CIFAR-10 |  | CIFAR-100 |  |
> |:---:|---|---|---|---|
> |  | s=2 | i.i.d. | s=10 | i.i.d. |
> | 1 | 72.0 | 74.0 | 43.3 | 43.2 |
> | 2 | 71.7 | 73.6 | 41.4 | 41.2 |
> | 3 | 72.0 | 73.1 | 37.3 | 37.4 |
> | 4 | 72.4 | 73.2 | 33.0 | 31.7 |
> | 5 | 73.0 | 73.2 | 28.6 | 26.6 |

---

### Official Review · Reviewer_JRVk · 2023-10-31

**Soundness:** 3 good
**Presentation:** 3 good
**Contribution:** 2 fair
**Rating:** 5
**Confidence:** 4

**Summary:**

This work proposed a novel lightweight unsupervised federated learning approach, FSTCBDG, to alleviate the computational and communication costs as well as the human labor of data annotations. The evaluation results illustrated the proposed FSTCBDG significantly outperforms the baseline methods.

**Strengths:**

1. Developed a lightweight unsupervised federated learning approach based on a single linear layer.

2. Designed a self-training objective for the linear classifier

3. Conducted extensive experiments to show the good performance of the proposed method over the baselines.

**Weaknesses:**

1. The class prototype augmentation based on Gaussian noise is not novel, since this idea has been used in prior work called PASS [Zhu CVPR 2021]. In addition, some follow-up works like [Zhu CVPR 2022] have pointed out that synthetic data from Gaussian-based augmentation would make some similar classes overlap with each other. Thus, the proposed method in this paper may not work well.

2. It may need to explain why the testing accuracy of baselines drops as the number of communication rounds increases.

3. Why did not compare with the FedUL baseline?

**References:**

[Zhu CVPR 2021] Prototype Augmentation and Self-Supervision for Incremental Learning, CVPR,2021

[Zhu CVPR 2022]. "Self-sustaining representation expansion for non-exemplar class-incremental learning." Proceedings of the IEEE/CVF Conference on Computer Vision and Pattern Recognition. 2022.

**Questions:**

Please see the comments above.

---

> ### Author Response · Authors · 2023-11-21
> **Reply from Authors**
>
> We thank the reviewer for dedicating time and effort to review our paper.
>
> **1. Novelty**
>
> - This paper introduces a novel setting of unsupervised federated learning and presents a lightweight framework designed to address this challenge effectively.
> - Notably, our work is the first to tackle federated learning with completely unlabeled data. While prior work, such as FedUL [1], claims to handle federated learning from unlabeled data, it necessitates knowledge of precise label frequencies for each client, which is often impractical in real-world applications.
> - Additionally, we propose a lightweight framework that incorporates the CLIP image encoder and text embeddings to enhance unsupervised federated learning. This framework trains a cleverly-initialized linear classifier on each client, significantly reducing computation and communication requirements.
> - As an illustrative solution within this framework, we introduce a resilient self-training approach with evolving pseudo-labels to address the unsupervised learning problem. To handle heterogeneous data distribution in federated learning, we propose a class-balanced data generation approach that generates synthetic feature instances from textual embeddings.
> - Experimental results demonstrate that our proposed method enhances the performance of CLIP's zero-shot prediction. Moreover, we show that our approach converges rapidly, substantially reducing the computation and communication demands on clients.
>
> **2. Class prototype augmentation**
>
> Different from the work [Zhu CVPR 2021] that generate Gaussian-based augmentation from prototypes during the model training, our method proposed to generate synthetic instances from the textual embeddings produced from the pre-trained CLIP text encoder. The textual feature vectors are trained to be discriminative such that the zero-shot prediction works. On the other hand, controlling the variance of the Gaussian distribution effectively avoids the feature distribution overlap across classes. We tried different $\sigma$ values and found that the simplest unit variance produces the best performance.
>
> **3. Reason that testing accuracy of baselines drops**
>
> As depicted in Figure 2 of our paper, a crucial observation regarding CLIP zero-shot prediction is that the predicted probability vectors exhibit high entropy, resembling a uniform distribution. When the CLIP image encoder is fixed, and the linear classifier is trained with strong supervision from ground-truth labels, it optimizes the classifier weights to diverge significantly from the textual embeddings. The divergence leads to biased and highly dynamic local classifiers, ultimately resulting in degraded performance. In contrast, our approach employs resilient self-training with evolving pseudo-labels to stabilize the local model updating process. Additionally, we incorporate class-balanced synthetic data generation to address the heterogeneous distribution of local data.
>
> **4. Comparing with FedUL baseline**
>
> As discussed in the related works section, it's crucial to highlight that FedUL necessitates precise knowledge of label frequencies for each client, making it an unfair comparison method for the proposed approach. Additionally, FedUL involves the transformation and recovery of model parameters, while the proposed lightweight framework maintains a fixed state for the image encoder. Consequently, FedUL cannot be seamlessly adapted to the proposed lightweight framework.

---

> > ### Comment · Reviewer_JRVk · 2023-11-22
> > **Thanks for your response**
> >
> > Thanks for your response! Regarding augmentation, I think you use the same idea as the PASS paper. PASS adds noise to the embeddings of images while you add noise to the embeddings of texts. In essence, the idea is quite similar. I think may you need to cite this work to respect the other authors. Thanks a lot!

---

> > > ### Author Response · Authors · 2023-11-22
> > > **Reply from Authors**
> > >
> > > We express our gratitude to the reviewer for providing additional feedback.
> > >
> > > As requested, we have incorporated the cited reference into the paper. It is important to reiterate that the innovation in our work does not lie in the use of the Gaussian augmentation method. Instead, our contribution centers around the introduction of a novel unsupervised federated learning setting and the proposal of a lightweight framework designed specifically for this context.
> > >
> > > It's worth noting that even the technique of feature-level Gaussian augmentation, which PASS employs, is not a novel introduction. Prior work, specifically by [DeVries and Taylor, 2017], had already employed this method for feature-level data augmentation. This technique is essentially a straightforward application of fundamental principles from the Gaussian distribution.
> > >
> > > Should there be any further inquiries or clarifications regarding the paper and the feedback received, please feel free to let us know.
> > >
> > > [DeVries and Taylor, 2017] DeVries, Terrance and Taylor, Graham W. Dataset augmentation in feature space. arXiv 2017.

---

### Official Review · Reviewer_b2QK · 2023-11-01

**Soundness:** 2 fair
**Presentation:** 2 fair
**Contribution:** 2 fair
**Rating:** 3
**Confidence:** 3

**Summary:**

This study leverages a pre-trained vision-language model for unsupervised image classification within a federated learning framework. The authors introduce two strategies to enhance the zero-shot prediction capabilities of CLIP. Experiments show that the proposed framework achieves better results compared to conventional supervised federated learning approaches.

**Strengths:**

1.	Compared to other federated learning scenarios, such as supervised and semi-supervised methods, unsupervised federated learning remains a relatively unexplored domain.
2.	From the ablation study, the proposed two approaches can effectively improve the zero-shot prediction accuracy of CLIP and mitigate the class imbalance problem to some extent.

**Weaknesses:**

1.	The contributions of this study did not meet the anticipated expectations. The two methods introduced lack novelty. The idea of refining pseudo-labels has been previously explored within the context of self-supervised learning. Additionally, the class-balanced data generation draws parallels with the Synthetic Minority Over-sampling Technique and can be categorized as an oversampling strategy.
2.	While the authors highlight the lightweight nature of their proposed framework, evidence throughout the paper seems insufficient. The sole indication of its lightweight character is the use of a linear classifier during training. However, this isn't a unique aspect as the baseline methods employ the same classifier. The purported lightweight advantage of the proposed framework isn't adequately substantiated. Furthermore, a comprehensive analysis of both computation and communication costs is essential to truly label the framework as lightweight and communication efficient.
3.	More experiments are required, especially for large scale datasets like ImageNet.
4.	There are some typos, like FedBR (Guo et al., 2023b) FedBR (Guo et al., 2023b), tranfer, etc.

**Questions:**

please respond to the weaknesses

---

> ### Author Response · Authors · 2023-11-21
> **Reply from Authors**
>
> We thank the reviewer for dedicating time and effort to review our paper.
>
> **1. Novelty of this paper**
> - This paper introduces a novel setting of unsupervised federated learning and presents a lightweight framework designed to address this challenge effectively.
> - Notably, our work is the first to tackle federated learning with completely unlabeled data. While prior work, such as FedUL (Lu et al, 2022), claims to handle federated learning from unlabeled data, it necessitates knowledge of precise label frequencies for each client, which is often impractical in real-world applications.
> - Additionally, we propose a lightweight framework that incorporates the CLIP image encoder and text embeddings to enhance unsupervised federated learning. This framework trains a cleverly-initialized linear classifier on each client, significantly reducing computation and communication requirements.
> - As an illustrative solution within this framework, we introduce a resilient self-training approach with evolving pseudo-labels to address the unsupervised learning problem. To handle heterogeneous data distribution in federated learning, we propose a class-balanced data generation approach that generates synthetic feature instances from textual embeddings.
> - Experimental results demonstrate that our proposed method enhances the performance of CLIP's zero-shot prediction. Moreover, we show that our approach converges rapidly, substantially reducing the computation and communication demands on clients.
>
> **2. Evidence of lightweight**
>
> This paper addresses the novel unsupervised federated learning problem and introduces a lightweight framework to address it. The lightweight nature of our approach is evident in the following aspects:
> - To minimize the number of parameters optimized in each client and transmitted between clients and the server, we introduce the CLIP image encoder, fixing it during local training. Only the weight parameters of the linear layer classifier are updated and transmitted.
> - To further reduce the number of communication rounds, we initialize the weight parameters of the linear layer classifier with text embeddings associated with class names. This results in a minimal number of local update epochs and communication rounds required for model convergence.
>
> A comprehensive analysis of both computation and communication costs is presented below. Comparing the training of the full model, including the image encoder and the classifier, with the ResNet-50 based CLIP image encoder and a linear layer classifier with 10 classes (e.g., CIFAR-10), the number of parameters in the former is approximately 38 million (M), while the latter is about 10 thousand (K), making a difference of almost 3800 times.
> - Computation Costs: Training the full model typically takes over 100 rounds with 5 local update epochs for each round according to FedAvg and FedNTD. This requires updating a total of $38M \times 100 \times 5$ parameters for each client. In contrast, our proposed lightweight method usually takes 10 rounds with 1 local update epoch for each round, involving the updating of only $10K \times 10 \times 1$ parameters for each client. The difference is approximately 190K times.
> - Communication Costs: Transmitting the full model incurs a cost of $38M \times 100 \times 32$ bits of bandwidth, while transmitting only the linear model costs $10K \times 10 \times 32$ bits of bandwidth. The difference is about 38K times.
>
> **3. Results on more datasets**
>
> We evaluate the proposed methods on three standard datasets: CIFAR-10, CIFAR-100, and CINIC-10, following established methods like FedAvg and FedNTD. These datasets cover various numbers of classes and are widely used for evaluating federated learning methods. While we acknowledge that including larger datasets would provide a more comprehensive evaluation, considering time limitations, we plan to explore this in future works.
>
> **4. Typos**
>
> We appreciate the reviewer for highlighting these typos, and they have been corrected in the latest revision.

---

### Meta-Review · Area_Chair_hBy4 · 2023-12-13

**Metareview:**

This paper proposes strategies to improve zero-shot performance of CLIP in image classification within a federated learning framework. The authors address the underexplored domain of unsupervised federated learning by combining different aspects of previous work with some novel elements. The showcased experiments are on small-scale datasets and reviewers raised some concerns about missing details and baselines. More extensive experiments on larger-scale datasets and extensive comparison to baselines would strengthen this paper.

**Justification For Why Not Higher Score:**

Small-scale and limited experimentation (e.g. CIFAR 10/100) and concerns about level of scientific contribution

**Justification For Why Not Lower Score:**

N/A

---

### Decision · Program_Chairs · 2024-01-16

Reject